# Spatial Characteristics and Implications of Grey Water Footprint of Major Food Crops in China

**Lin Wang [1,2,\*], Yutong Zhang [2], Ling Jia [1,2], Guiyu Yang [1,2], Yizhen Yao [2] and Weiping Wang [3]**

[1]   State Key Laboratory of Simulation and Regulation of Water Cycle in River Basin, China Institute of Water Resources and Hydropower Research, Beijing 100038, China; jialing@iwhr.com (L.J.); guiyuy@iwhr.com (G.Y.)

[2]   Department of Water Resources, China Institute of Water Resources and Hydropower Research, Beijing  100038, China; zyt1169742688@hotmail.com (Y.Z.); yaoyz2019@hotmail.com (Y.Y.)

[3]   Division of Personnel, Labor and Education, China Institute of Water Resources and Hydropower Research, Beijing  100038, China; wpwang@iwhr.com

\*   Correspondence: wanglin@iwhr.com; Tel.: +86-10-6878-5512

**Abstract:** The estimated, effective increase of agricultural fertilizer applied in China by 10.57 Mts from 2006 to 2016 is a crucial factor affecting the water environment.  Based on analyzing the nitrate-leaching rate, the nitrogen-fertilizer application rate, and crop yield in wheat and maize key cultivation divisions in China, this paper applied the grey water footprint analytical method to estimate THE grey water footprint and its proportion to total water footprint and analyzed the spatial differences from 2012 to 2016. Results showed that the grey water footprint of wheat was higher in North and Northwest China with an increasing trend, while that of maize was higher in Southwest and Northwest China because of high nitrogen application rates and low yields in these regions. Except for the Southwestern division, wheat's grey water footprint was about 1.3 times higher than the blue water footprint, while, for maize, it was two to three times higher. When analyzing and planning water demand for crop irrigation, the water required for nonpoint source pollution due to chemical fertilizers should be considered. Focusing blue water (irrigation) alone, while neglecting green water and ignoring grey water footprints, it might lead to overestimation of available agricultural water resources and failure to meet the goals of sustainable use of water resources.

**Keywords:** grey water footprint; food crops; nitrogen application rate; nitrate-nitrogen leaching rate

## 1. Introduction

The "1st National Census on Pollution Sources Bulletin" by the China Ministry of Environment Protection [1], showed that, in China, agricultural production-related (including cultivation, livestock, and poultry farming and aquaculture) emissions of major pollutants, such as chemical oxygen demand, total nitrogen, and total phosphorus, have exceeded industrial and human activity pollution levels. The  oxygen, nitrogen, and phosphorus pollutants have become the main source of pollution in the country, which accounts for 43.7%, 57.2%, and 67.4% of the total emissions, respectively. From 2006 to 2016, the total amount of agricultural fertilizer applied (i.e., effective amount of nitrogen-containing fertilizer applied nationwide), increased by 10.57 million tons [2], which shows a marked increasing trend, and greatly affects the aquatic environment quality and ecological environment health. Therefore, it is fundamental to study the spatial distribution of agricultural nonpoint source (NPS) pollution generation and emission, and objectively analyze the effect of agricultural production on aquatic environments to generate prevention strategies for the non-point source pollution and to control and reduce agricultural pollution.  In this study, the grey water footprint (WFgrey) methodology was

used to analyze the relationship between water quality and water quantity, and to measure directly the effect of the application of crop fertilizers on the utilization and quality of freshwater resources. In 2008, Hoekstra and Chapagain first proposed the concept of WFgrey, which was later elaborated in the Grey Water Footprint Guidelines [3]. WFgrey is defined by Hoekstra et al. [4,5] as the volume of freshwater required to assimilate a pollutant to a specific environmental water quality standard under the current water quality standards. WFgrey for crop production is the amount of water required to assimilate the pollutants produced by crop yield per unit area to bring the water used for crop irrigation to the local environmental discharge standards [6]. Many studies have been carried out in China and other countries since this methodology was introduced. Studies from other countries focused on the evaluation of agricultural WFgrey at regional and basin scales [7,8]. For example, Simona-Andreea et al. [7] calculated the WFgrey for eight major crops in the Prut-Barlad catchment from 2005 to 2008. Their results showed that wheat had the highest WFgrey (58 Mm$^3$/year), which is followed by vegetables (52 Mm$^3$/year). Mekonnen et al. [8] summarized the agricultural, industrial, and domestic activity WFgrey, and water pollution related to nitrogen loads, to fresh water in the basin area of different countries from 2002 to 2010. Results showed that China had the highest nitrogen emissions, which accounts for about 45% of the global grand total and results in the following WFgrey ranking: agriculture > domestic activity > industry. Grain crops have the highest WFgrey caused by nitrogen loads among crops, which is followed by vegetables. Domestic WFgrey studies focused on the calculation and evaluation of agricultural WFgrey caused by pollution of chemical fertilizers and pesticides at regional scales [9,10]. Zhang et al. [9] used fixed leaching rates to calculate the WFgrey of the winter wheat-summer maize rotation, caused by the nitrogen fertilizer application in the North China Plain from 1986 to 2010, and analyzed the temporal and spatial differences. They concluded that WFgrey for production has a fluctuating and increasing trend every year. Cao et al. [10] conducted a quantitative evaluation of WFgrey of grain production in the Hetao Irrigation district of Inner Mongolia from 2005 to 2008, from the perspectives of NPS pollution and salinization. Ranking of WFgrey for the NPS pollution was nitrogen fertilizer > pesticide > phosphorus fertilizer. The salinization footprint (SP, salt accumulation and groundwater salinity) accounted for the largest proportion of the total WFgrey, while total WFgrey accounted for less than 10% of the total water footprint (WFtot), with a decreasing trend every year. The above studies were only for a certain region in China, and the leaching rate was fixed, while spatial heterogeneity was not considered.

This study systemically recorded and summarized the nitrate-nitrogen leaching rate, the nitrogen-fertilizer application rate, and crop yield per unit area in key wheat and maize cultivation divisions of China. The WFgrey methodology was applied to estimate the WFgrey and its proportion to the WFtot in those divisions from 2012 to 2016. Spatial differences were analyzed using the provincial administrative region as a basic unit. Our objectives were to generate useful information leading to (1) gain a better understanding of the effect of food crop production on freshwater resources, (2) provide technical support for agricultural water planning, and (3) prevent and control of NPS pollution.

## 2. Materials and Methods

### 2.1. Grey Water Footprint and Total Water Footprint of Food Crops

The calculation of the grey water footprint for crop production (WF$_{proc,grey}$ m$^3$/kg) was done as described in "The Water Footprint Assessment Manual" by Hoekstra et al. [5], using the following equation.

$$\text{WF}_{\text{proc,grey}} = \frac{(\alpha \times \text{AR})/(\text{C}_{\text{max}} - \text{C}_{\text{nat}})}{\text{Y}} \tag{1}$$

where AR is the amount of fertilizer applied per hectare (kg/ha), $\alpha$ is the leaching rate, which represents the proportion of pollutants entering the water body to the total amount of fertilizer, C$_{max}$ is the maximum allowable concentration (kg/m$^3$), C$_{nat}$ is the natural background concentration of pollutants (kg/m$^3$), and Y is the yield per hectare (kg/ha).

Agricultural fertilizers include nitrogen, phosphate, potassium, and compound fertilizers. Nitrogen is one of the most important nutrient elements in agricultural ecosystems [11]. The main compounds of nitrogen migration in farmlands are $NO_3^-$, $NO_2^-$, and $NH_4^+$. According to the National Bureau of Statistics, in 2015, the analyzed effective amount of agricultural nitrogen-fertilizer applied was 43,900 tons, which accounts for 46.1% of the total effective amount of fertilizer applied [2], while only around 35% of the nitrogen applied to the soil was absorbed and utilized by the crop. The rest of the applied nitrogen remained in the soil or entered the environment through ammonia volatilization, nitrification and denitrification, runoff, and leaching. Nitrate-nitrogen ($NO_3^-$) is the most common and active form of soil nitrogen transformation and migration. The negatively charged $NO_3^-$ cannot be easily adsorbed by soil colloids, and it is easily leached into the soil, where it just as easy contaminates shallow groundwater reservoirs [12]. This is the main source of nitrogen pollution by nitrate in shallow groundwater [13]. Therefore, this study considered nitrogen-containing fertilizers (nitrogen fertilizer, compound fertilizer) as the research objects of pollution and used the nitrate-nitrogen leaching rate as the parameter to calculate $WF_{grey}$.

The total amount of freshwater consumed and contaminated during crop growth is the total water footprint ($WF_{tot}$) for crop production, which is numerically the sum of green ($WF_{green}$), blue ($WF_{blue}$), and grey ($WF_{grey}$) water footprints in which the $WF_{green}$ and $WF_{blue}$ represents "the total rainwater evaporated from the field during the growing period" and "the total irrigation water evaporated from the field" [5] per unit crop yield, while $WF_{grey}$, mentioned above, is often ignored.

## 2.2. Parameter Selection and Data Source

### 2.2.1. Nitrate-Nitrogen Leaching Rate

Nitrogen leaching loss refers to the process whereby the remaining nitrogen that is not absorbed by plant roots infiltrates into deep soil and groundwater carried by rainfall or irrigation water, and is discharged into rivers and lakes through ditches, which leads to nitrogen loss in farmlands [14]. The Nitrate-N leaching rate is the ratio of nitrate-nitrogen leaching loss to the nitrogen application rate. This study retrieved and filtrated 76 experimental data sets on the nitrate-nitrogen leaching rate in the key wheat and maize cultivation divisions of China through the review of the literature [15–32]. These data appear to be a positive relation between the nitrogen application rate and the leaching rate of wheat and maize (Figure 1). Summarized results for wheat and maize are listed in Tables 1 and 2, respectively. As the range of the leaching rate is wide, the average nitrate-nitrogen leaching rate from the relevant provincial administrative regions of each division is used to represent the overall situation of the division.

According to the experimental data on the nitrate-nitrogen leaching rate in China, the nitrate-nitrogen leaching rate of wheat is the highest (~19.5%) in North China, which is followed by Northwest China (~12.3%), East China (~6.6%), and Southwest China (~1.9%). The nitrate-nitrogen leaching rate of maize is the highest in Northeast China (~19.3%), which is followed by North China, (~16.4%), Southwest China (~13.8%), Northwest China (~12.5%), and East China (~6.8%).

**Table 1.** Nitrogen application rate and nitrate-nitrogen leaching rate in key wheat cultivation divisions.

| Division | Location | N Application Rate (kg/ha) | Nitrate-N Leaching Loss Rate (kg/ha) | Leaching Rate (%) | Reference |
|---|---|---|---|---|---|
| North China | Chinese Agricultural University campus test field | 120 | 22 | 18.3 | [15] |
| | | 240 | 117 | 48.8 | |
| | | 360 | 166 | 46.1 | |
| | | 120 | 1 | 0.8 | |
| | | 240 | 62 | 25.8 | |
| | | 360 | 131 | 36.4 | |
| | The piedmont region of Taihang Mountains | 200 | 22 | 11.0 | [16] |
| | | 400 | 110 | 27.5 | |
| | Quzhou County, Hebei Province | 258.8 | 23.1 | 8.9 | [17] |
| | | 341.3 | 16.5 | 4.8 | |
| | | 354.9 | 14.4 | 4.1 | |
| | | 303.6 | 4.2 | 1.4 | |
| | Average | — | — | 19.5 | — |
| East China | Taihu Plains | 225 | 11.7 | 5.2 | [18] |
| | | 300 | 13.2 | 4.4 | |
| | Changshu, Jiangsu | 150 | 5.52 | 3.7 | [19] |
| | | 225 | 14.71 | 6.5 | |
| | | 300 | 31.57 | 10.5 | |
| | | 450 | 40.69 | 9.0 | |
| | Average | — | — | 6.6 | — |
| Southwest China | Chongqing | 75 | 1.67 | 2.2 | [20] |
| | | 150 | 1.98 | 1.3 | |
| | | 225 | 2.26 | 1.0 | |
| | Chongqing | 150 | 4.42 | 3.0 | [21] |
| | Average | — | — | 1.9 | — |
| Northwest China | Guanzhong Plain | 250 | 6.3 | 2.5 | [22] |
| | Yangling, Shaanxi | 250 | 4.67 | 1.9 | [23] |
| | Yangling, Shaanxi | 250 | 77.5 | 31.0 | [24] |
| | Hanzhong, Shaanxi | 250 | 37.5 | 15.0 | [24] |
| | Yangling, Shaanxi | 250 | 5.22 | 2.1 | [25] |
| | Wuwei, Gansu | 225 | 47.49 | 21.1 | [26] |
| | Average | — | — | 12.3 | — |

Note: "—" indicates no data available.

**Table 2.** Nitrogen application rate and nitrate-nitrogen leaching rate in key maize cultivation divisions.

| Division | Location | N Application Rate (kg/ha) | Nitrate-N Leaching Loss Rate (kg/ha) | Leaching Rate (%) | Reference |
|---|---|---|---|---|---|
| North China | Chinese Agricultural University campus test field | 120 | 21 | 17.5 | [15] |
| | | 240 | 38 | 15.8 | |
| | | 360 | 16 | 4.4 | |
| | | 120 | 70 | 58.3 | |
| | | 240 | 104 | 43.3 | |
| | | 360 | 197 | 54.7 | |
| | Bayannur, Inner Mongolia | 80 | 5.41 | 6.8 | [27] |
| | | 160 | 12.9 | 8.1 | |
| | | 240 | 14.36 | 6.0 | |
| | | 320 | 19.85 | 6.2 | |
| | Quzhou County, Hebei Province | 186.3 | 9.3 | 5.0 | [17] |
| | | 148.3 | 11.6 | 7.8 | |
| | | 151.8 | 2.5 | 1.7 | |
| | | 158.7 | 1.2 | 0.8 | |
| | Average | — | — | 16.9 | — |
| East China | Shandong Agricultural University | 240 | 20.44 | 8.5 | [28] |
| | | 480 | 24.54 | 5.1 | |
| | Average | — | — | 6.8 | — |
| Southwest China | Yanting County | 150 | 22.18 | 14.8 | [29] |
| | | 150 | 19.25 | 12.8 | |
| | Average | — | — | 13.8 | — |
| Northwest China | Guanzhong Plain | 250 | 6.3 | 2.5 | [22] |
| | Yangling, Shaanxi | 250 | 4.67 | 1.9 | [23] |
| | Mizhi County, North Shaanxi | 250 | 102.5 | 41.0 | [24] |
| | Yangling, Shaanxi | 250 | 5.22 | 2.1 | [25] |
| | Wuwei, Gansu | 225 | 54.16 | 24.1 | [26] |
| | Gansu Agricultural University | 150 | 4.64 | 3.1 | [30] |
| | | 300 | 4 | 1.3 | |
| | Average | — | — | 10.9 | — |
| Northeast China | Linghai City | 210 | 82.2 | 39.1 | [31] |
| | | 263 | 99.6 | 37.9 | |
| | Northeast Agricultural University | 229.42 | 0.16 | 0.1 | [32] |
| | | 290.66 | 0.21 | 0.1 | |
| | Average | — | — | 19.3 | — |

Note: "—" indicates no data available.

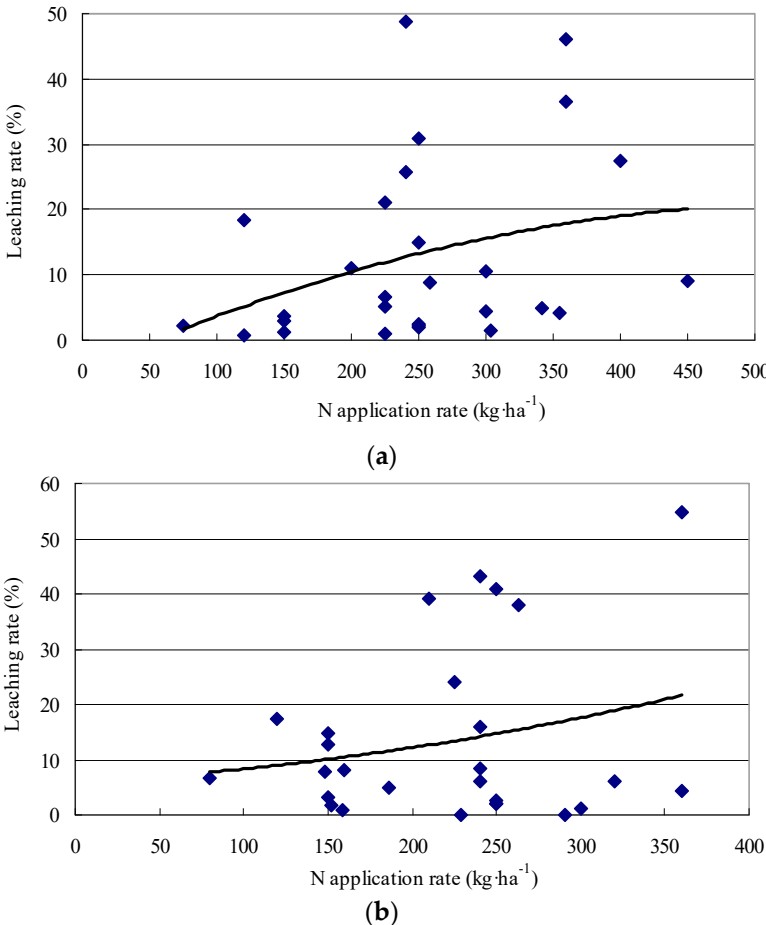

**(a)**

**(b)**

**Figure 1.** Relation between the nitrogen application rate and the leaching rate of wheat (**a**) and maize (**b**).

2.2.2. Average Application Rate of Nitrogen-Containing Fertilizers AR

Nitrogen-containing fertilizers include nitrogen fertilizers and compound fertilizers. Surveys indicate that approximately one-third of the farmers in China use compound fertilizers containing total nitrogen, available phosphorus pentoxide and potassium oxide in 15%, 15% and 15% percentage composition [33]. Therefore, the nitrogen content of the compound fertilizer is computed at 15%. The average application rate of nitrogen-containing fertilizer was calculated using the following equation.

$$AR = AR_N + AR_{NPK} \times 0.15 \tag{2}$$

where AR is the analyzed effective application rate of nitrogen-containing fertilizer on a per unit area (kg/ha) basis, $AR_N$ is the analyzed effective average application rate of nitrogen fertilizer per acre (kg/ha), and $AR_{NPK}$ is the analyzed effective average application rate of compound fertilizer per acre (kg/ha).

The analyzed effective average application rate of nitrogen-containing fertilizer per unit area is derived from the "National Agricultural Products Cost-Revenue Data Compilation 2012–2016" by the China National Development and Reform Commission [34]. Provinces that lacked the average application rate of nitrogen-containing fertilizer data for wheat and maize were assigned the average of other provinces in the same division.

From Table 3, it can be understood that, from 2012 to 2016, the nitrogen application rate in most regions of China trended higher except those of wheat and maize in Hebei, Shandong, and maize in Northeast China.

**Table 3.** Nitrogen application rate from 2012 to 2016.

| Division | Provincial Administrative Region | Wheat(kg/ha) | | | | | Maize(kg/ha) | | | | | Ratio | |
|---|---|---|---|---|---|---|---|---|---|---|---|---|---|
| | | 2012 (1) | 2013 (2) | 2014 (3) | 2015 (4) | 2016 (5) | 2012 (6) | 2013 (7) | 2014 (8) | 2015 (9) | 2016 (10) | Wheat (5)/(1) | Maize (10)/(6) |
| North China | Beijing | 240.3 | 238.2 | 242.4 | 242.4 | 251.0 | 151.0 | 147.4 | 153.4 | 148.0 | 158.2 | 1.04 | 1.05 |
| | Tianjing | 240.3 | 238.2 | 242.4 | 242.4 | 251.0 | 151.0 | 147.4 | 153.4 | 148.0 | 158.2 | 1.04 | 1.05 |
| | Hebei | 218.2 | 212.5 | 211.8 | 216.6 | 214.6 | 127.9 | 126.5 | 126.9 | 126.0 | 117.2 | 0.98 | 0.92 |
| | Inner Mongolia | 312.4 | 323.4 | 329.1 | 333.8 | 355.6 | 182.8 | 174.9 | 179.2 | 173.7 | 203.7 | 1.14 | 1.11 |
| | Shanxi | 190.4 | 178.6 | 186.2 | 176.8 | 182.8 | 142.3 | 140.8 | 154.1 | 144.4 | 153.7 | 0.96 | 1.08 |
| East China | Shandong | 163.3 | 164.2 | 167.2 | 164.4 | 156.4 | 149.7 | 168.5 | 165.7 | 147.1 | 141.5 | 0.96 | 0.94 |
| | Shanghai | 175.4 | 176.6 | 184.7 | 177.6 | 177.1 | 190.9 | 198.8 | 207.2 | 186.5 | 181.2 | 1.01 | 0.95 |
| | Zhejiang | 175.4 | 176.6 | 184.7 | 177.6 | 177.1 | 190.9 | 198.8 | 207.2 | 186.5 | 181.2 | 1.01 | 0.95 |
| | Jiangsu | 216.1 | 218.8 | 227.9 | 226.9 | 228.5 | 248.1 | 228.9 | 255.3 | 228.8 | 236.5 | 1.06 | 0.95 |
| | Jiangsu | 175.4 | 176.6 | 184.7 | 177.6 | 177.1 | 190.9 | 198.8 | 207.2 | 186.5 | 181.2 | 1.01 | 0.95 |
| | Fujian | 175.4 | 176.6 | 184.7 | 177.6 | 177.1 | 190.9 | 198.8 | 207.2 | 186.5 | 181.2 | 1.01 | 0.95 |
| | Anhui | 146.9 | 146.9 | 158.9 | 141.5 | 146.4 | 175.1 | 199.1 | 200.7 | 183.5 | 165.6 | 1.00 | 0.95 |
| Southwest China | Sichuan | 93.0 | 90.6 | 104.6 | 99.9 | 104.6 | 163.8 | 159.7 | 155.7 | 158.4 | 157.7 | 1.12 | 0.96 |
| | Chongqing | 112.0 | 109.9 | 120.2 | 119.4 | 131.7 | 221.7 | 189.7 | 193.8 | 187.8 | 186.1 | 1.18 | 0.84 |
| | Guizhou | 112.0 | 109.9 | 120.2 | 119.4 | 131.7 | 186.8 | 198.3 | 202.3 | 230.6 | 219.7 | 1.18 | 1.18 |
| | Tibet | 112.0 | 109.9 | 120.2 | 119.4 | 131.7 | 209.9 | 207.3 | 207.8 | 213.2 | 211.7 | 1.18 | 1.01 |
| | Yunnan | 131.1 | 129.3 | 135.7 | 138.8 | 158.7 | 267.4 | 281.3 | 279.3 | 276.0 | 283.3 | 1.21 | 1.06 |
| Northwest China | Shaanxi | 229.1 | 201.0 | 206.2 | 217.1 | 202.4 | 263.6 | 255.0 | 273.3 | 245.9 | 267.7 | 0.88 | 1.02 |
| | Xinjiang | 222.0 | 228.0 | 236.5 | 247.6 | 249.5 | 259.4 | 263.9 | 258.6 | 262.6 | 270.8 | 1.12 | 1.04 |
| | Gansu | 173.9 | 163.9 | 200.6 | 189.7 | 178.2 | 262.7 | 252.2 | 265.3 | 272.6 | 273.6 | 1.02 | 1.04 |
| | Ningxia | 194.6 | 220.8 | 211.6 | 220.6 | 228.2 | 234.0 | 243.7 | 253.9 | 291.4 | 259.1 | 1.17 | 1.11 |
| | Qinghai | 204.9 | 203.4 | 213.7 | 218.8 | 214.6 | 254.9 | 253.7 | 262.8 | 268.1 | 267.8 | 1.05 | 1.05 |
| Northeast China | Heilongjiang | — | — | — | — | — | 114.9 | 125.2 | 133.3 | 130.9 | 139.0 | — | 1.21 |
| | Jilin | — | — | — | — | — | 136.7 | 129.2 | 115.6 | 104.3 | 109.0 | — | 0.80 |
| | Liaoning | — | — | — | — | — | 157.2 | 155.8 | 151.3 | 124.1 | 128.1 | — | 0.82 |

Note: "—" indicates no data available.

### 2.2.3. Other Parameters

The yields per unit area of each provincial administrative region in the key divisions were obtained from the national data of the National Bureau of Statistics of the People's Republic of China (http://data.stats.gov.cn/index.htm). The standard concentration limit of nitrate (counts as Nitrogen), expressed as $C_{max} = 0.01$ kg/m³, in the "Surface Water Environmental Quality Standard" (GB 3838-2002) [35] was used as a discharge standard for pollutants. The natural background concentration of the receiving water body is currently unavailable, and the estimated value is low. Therefore, it is simply treated, according to Hoekstra et al. [5].

## 3. Results

### 3.1. Nitrogen Application Rate Per Unit Area and Yield Spatial Distribution

Wheat. From 2012 to 2016, the nitrogen application rate in wheat showed an increasing trend in North China, Southwest China, and Northwest China, whereas, in East China, it remained relatively unchanged. The five-year averages ranked as follows: North China 242 > Northwest China 211 > East China 178 > Southwest 118 kg/ha. The high average of the North China division was affected by the high nitrogen application rate prevalent in Inner Mongolia (330 kg/ha). Next, Northwest and East China were affected by higher nitrogen application rates in Xinjiang and Jiangsu, respectively. As shown in Figure 2a, when the high nitrogen application rates in Inner Mongolia, Xinjiang, and Jiangsu were excluded, the average nitrogen-application rate in the three regions was not much different. The wheat yield per unit area showed an increasing trend from 2012 to 2016. The five-year averages ranked as follows: North China 4653 > East China 4234 > Northwest China 3970 > Southwest China 3501 kg/ha. The yield per unit area varied greatly in different provinces within each division. At the provincial region administrative level, Tibet, Shandong, Hebei, Xinjiang, and Anhui showed a higher yield per unit area, but there was no correlation between the yield per unit area and the nitrogen application (Figure 2b).

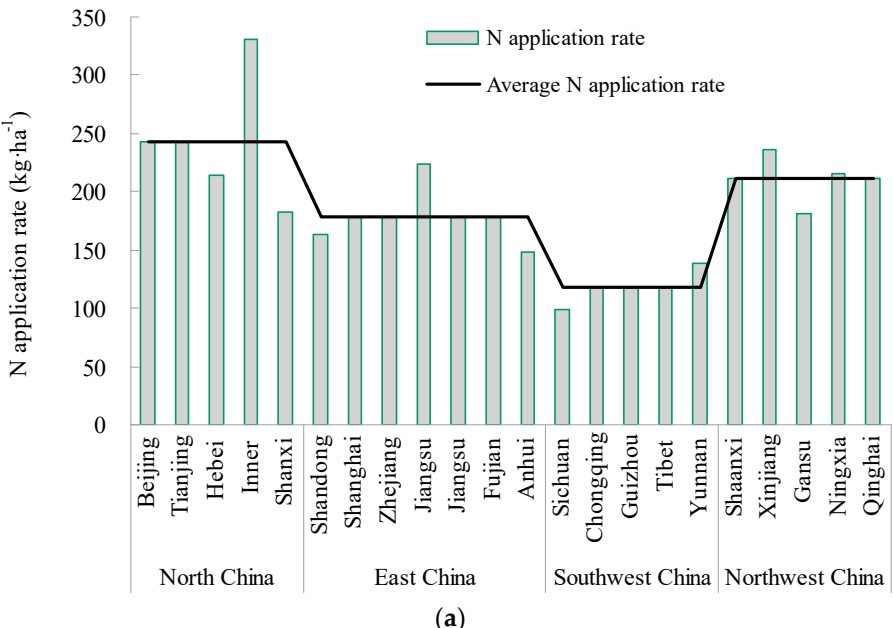

(a)

**Figure 2.** *Cont.*

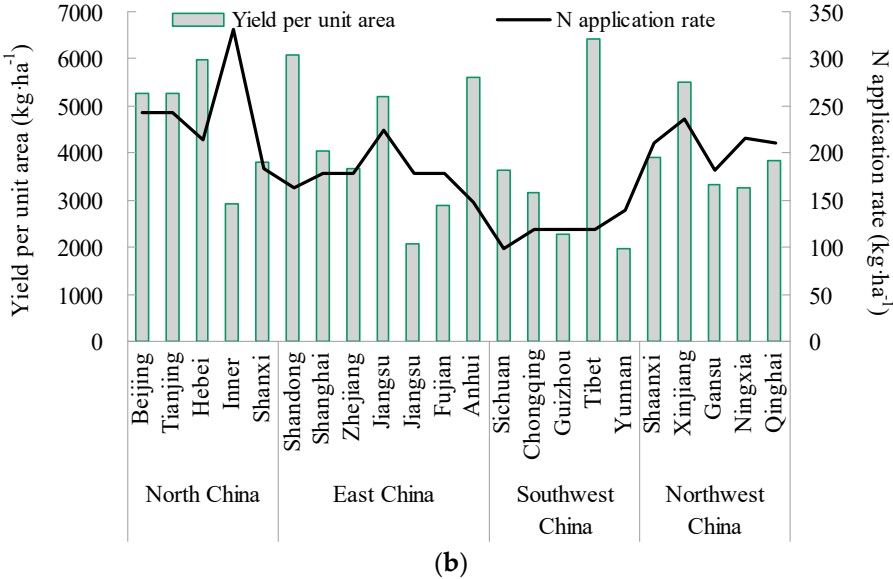

**Figure 2.** Nitrogen application rate, yield per unit area, and average value of wheat in each province from 2012 to 2016.

Maize. From 2012 to 2016, the nitrogen application rate in maize showed an increasing trend in North China, Southwest China, and Northwest China, while, in East China, it first increased and then decreased. During the same period, in Northeast China, it showed a decreasing trend. The five-year averages ranked as follows: Northwest 261 > Southwest China 209 > East China 192 > North China 151 > Northeast China 130 kg/ha. When the high nitrogen application rates from Inner Mongolia in North China, Jiangsu in East China, and Yunnan in Southwest China were excluded, the ranking of the average nitrogen application rate in the five divisions remained the same, as shown in Figure 3a, which indicates that there are significant regional differences in the nitrogen application rate for maize production, especially in the Northwest. From 2012 to 2016, maize yield per unit area showed an increasing trend in North, Southwest, and Northeast China, while it remained stable in East China and it showed a decreasing trend in Northwest China. The five-year averages ranked as follows: Northeast 6582 > Northwest China 6461.67 > North China 5775 > East China 5225 > Southwest China 5173 kg/ha. Maize yield per unit area was low in East and Southwest China. Jilin in Northeast China, and Ningxia, Xinjiang, and Qinghai in Northwest China had higher yield per unit area while, in other provincial administrative regions, it ranged between 4100 and 6600 kg/ha. There was no correlation between the yield per unit area and the nitrogen application rate (Figure 3b).

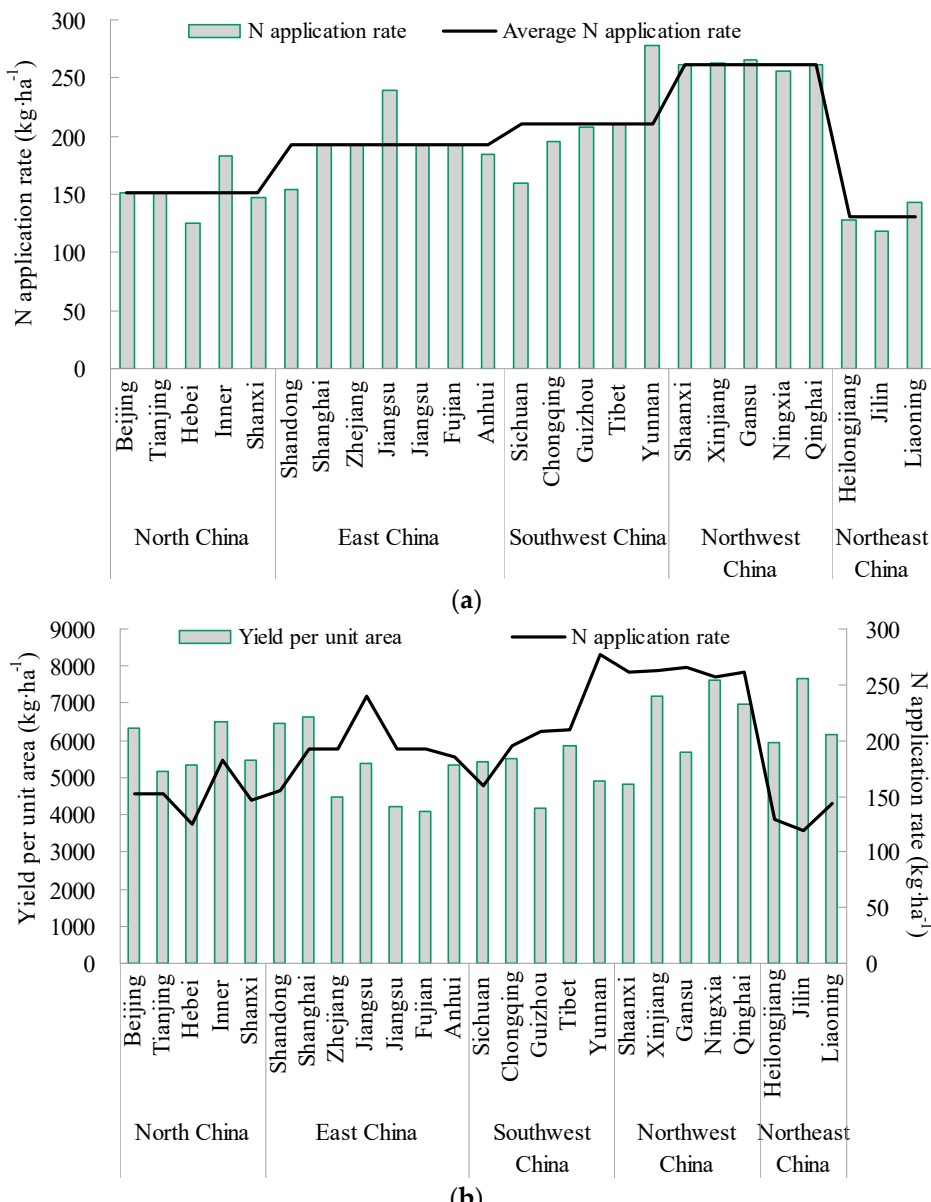

**Figure 3.** Nitrogen application rate, yield per unit area, and average value of maize in each province from 2012 to 2016.

*3.2. Grey Water Footprint and Its Spatial Distribution*

The WF$_{grey}$ of wheat and maize in each provincial administrative region from 2012 to 2016 were calculated using Equation (1). In general, there was a significant divisional difference on the WF$_{grey}$ of wheat. The change within each division was not significant, except for the North China division. From 2012 to 2016, North China showed an increasing trend for WF$_{grey}$, which rose from 1.11 to 1.16 m$^3$/kg, which is a 4.5% increase. The Northwest division showed an initial increase followed by a decrease, from 0.71 m$^3$/kg to 0.73 m$^3$/kg, which is a 4.2% net increase in five years. East China and Southwest China were relatively stable. Regarding the average values of WF$_{grey}$ of wheat and maize in each region in the five years, Figure 4 shows the difference between grey water footprints for wheat and maize in spatial distribution. WF$_{grey}$ ranked as follows: North China 1.13 > Northwest China 0.74 > East China 0.29 > Southwest China 0.09 m$^3$/kg. There was a significant spatial difference, within which Inner Mongolia ranked highest at 2.21 m$^3$/kg.

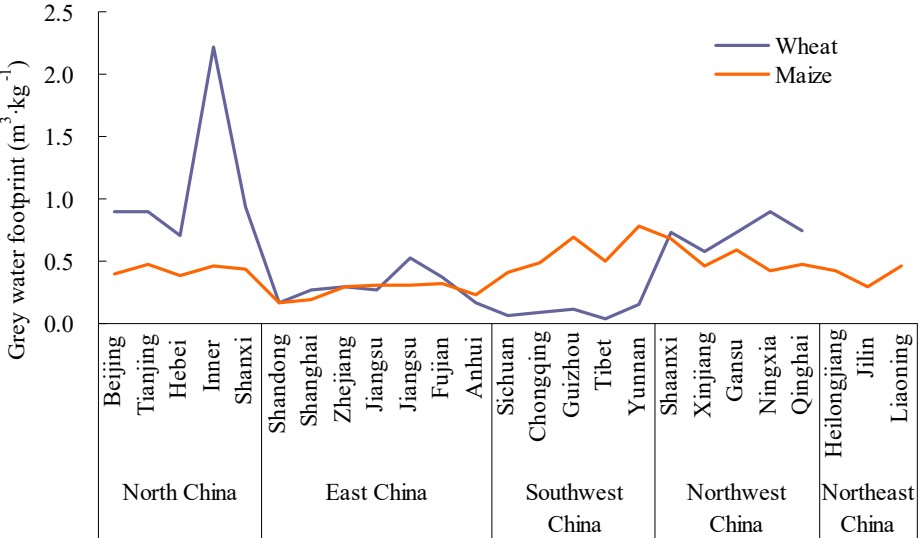

**Figure 4.** Five-year average of the grey water footprint for wheat and maize production in each province.

Compared with wheat, the divisional difference in $WF_{grey}$ of maize was small even though there were fluctuations within each division. From 2012 to 2016, North China, Northeast China, East China, and Southwest China showed a trend in which it initially increased and then decreased. The $WF_{grey}$ of maize increased by 2.3%, decreased by 8.1%, decreased by 4.2%, and decreased by 1.8% in these four divisions, respectively, when 2006 data were compared to data from 2012. Northwest China showed a trend of decreasing $WF_{grey}$ first and then increasing, for a net 5.6% gain. The five-year averages of maize $WF_{grey}$ ranked as follows: Southwest China 0.56 > Northwest China 0.54 > North China 0.44 > Northeast China 0.37 > East China 0.24. Within this range, Yunnan showed the highest value, at 0.78 m$^3$/kg.

When comparing the average nitrogen application rate, the nitrate-nitrogen leaching rate, the yield per unit area, and the $WF_{grey}$ of wheat in each district to the corresponding key cultivation divisional averages, it shows a higher $WF_{grey}$ of wheat than the key divisional average in North and Northwest China but lower than the key divisional average in East China and Southwest China, as shown in Figure 5a. On the other hand, the $WF_{grey}$ of maize was high in Southwest China and Northwest China, low in East and Northeast China, and average in North China, as shown in Figure 5b.

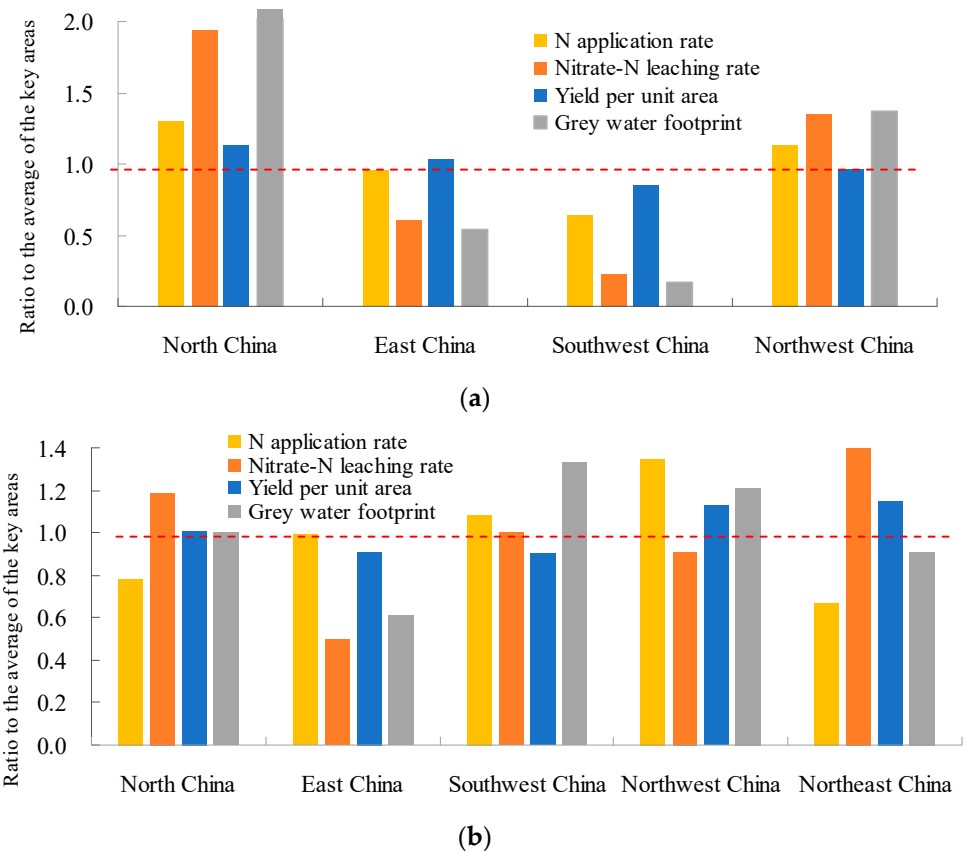

**Figure 5.** Ratio of WFgrey of wheat (**a**) and maize (**b**) in each division to the key divisional average, and the factors affecting it.

### 3.3. The Proportion of the Total Water Footprint and the Grey Water Footprint

To analyze the effect of $WF_{grey}$ on $WF_{tot}$, this study extracted the utilization of natural precipitation (green water) and the demand of irrigation water (blue water) during the growth period of wheat and maize from the "National Water Resources Integrated Planning" Irrigation Water Demand Inquiry System 4.0. Considering the inter-annual precipitation and climate change, the average value from 1956 to 2000 was used to reflect the multi-year average, which was then divided by the average of yield per unit area of wheat and maize in the past five years, from 2012 to 2016. This was completed to obtain the green and blue water footprints of the two main food crops on the current irrigation division and yield capacity. The composition of $WF_{tot}$ for wheat and maize is shown in Figure 6. The proportions of $WF_{green}$, $WF_{blue}$, and $WF_{grey}$ for wheat and maize from 2012 to 2016 are listed in Table 4.

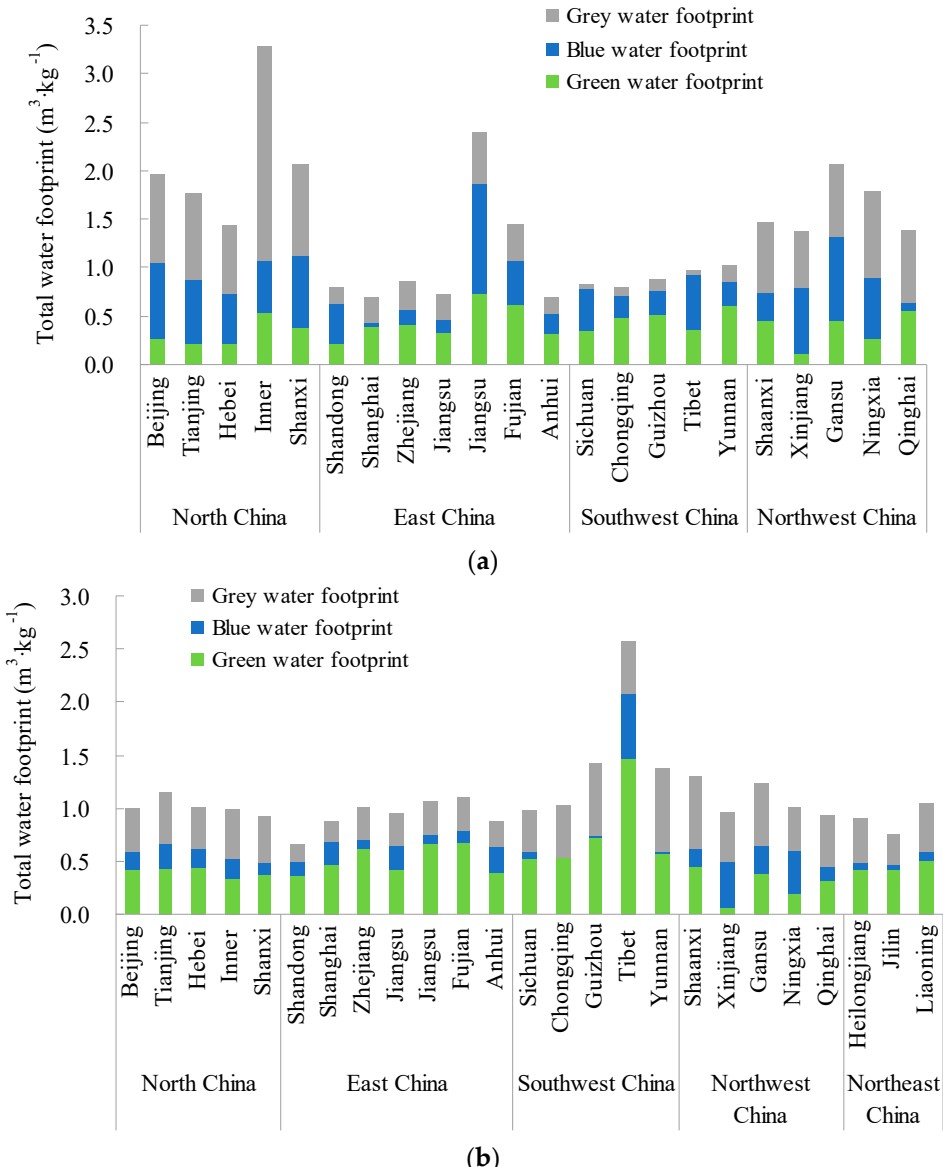

**Figure 6.** Mean value of the total water footprint of wheat (**a**) and maize (**b**) from 2012 to 2016.

**Table 4.** The proportion of green water, blue water, and the grey water footprint of wheat and maize from 2012 to 2016.

| Division | Provincial Administrative Region | Wheat | | | Maize | | |
|---|---|---|---|---|---|---|---|
| | | WF_blue (%) | WF_green (%) | WF_grey (%) | WF_blue (%) | WF_green (%) | WF_grey (%) |
| North China | Beijing | 40.6 | 13.5 | 45.9 | 17.4 | 42.7 | 39.9 |
| | Tianjin | 37.0 | 12.3 | 50.7 | 19.0 | 38.8 | 42.2 |
| | Hebei | 36.1 | 15.0 | 48.9 | 18.4 | 43.5 | 38.1 |
| | Inner Mongolia | 16.3 | 16.2 | 67.5 | 20.0 | 33.6 | 46.4 |
| | Shanxi | 35.9 | 18.7 | 45.4 | 11.2 | 41.1 | 47.7 |
| East China | Shandong | 52.9 | 26.5 | 20.6 | 20.9 | 54.5 | 24.6 |
| | Shanghai | 4.8 | 56.4 | 38.9 | 24.0 | 53.3 | 22.7 |
| | Zhejiang | 16.3 | 48.6 | 35.0 | 9.7 | 60.9 | 29.4 |
| | Jiangsu | 18.9 | 44.8 | 36.3 | 23.3 | 44.7 | 32.1 |
| | Jiangxi | 47.6 | 30.5 | 21.9 | 8.8 | 62.0 | 29.1 |
| | Fujian | 32.5 | 41.6 | 25.9 | 9.7 | 61.2 | 29.1 |
| | Anhui | 31.2 | 45.6 | 23.2 | 26.9 | 45.9 | 27.3 |

**Table 4.** *Cont.*

| Division | Provincial Administrative Region | Wheat | | | Maize | | |
|---|---|---|---|---|---|---|---|
| | | $WF_{blue}$ (%) | $WF_{green}$ (%) | $WF_{grey}$ (%) | $WF_{blue}$ (%) | $WF_{green}$ (%) | $WF_{grey}$ (%) |
| Southwest China | Sichuan | 50.8 | 41.9 | 7.3 | 5.1 | 53.9 | 41.0 |
| | Chongqing | 28.8 | 60.7 | 10.5 | 0.6 | 51.5 | 47.9 |
| | Guizhou | 28.2 | 58.6 | 13.2 | 1.0 | 50.6 | 48.4 |
| | Tibet | 58.3 | 37.5 | 4.3 | 24.1 | 56.6 | 19.3 |
| | Yunnan | 25.2 | 59.5 | 15.3 | 1.2 | 41.5 | 57.3 |
| Northwest China | Shaanxi | 20.1 | 30.2 | 49.7 | 12.6 | 35.1 | 52.3 |
| | Xinjiang | 49.8 | 7.8 | 42.4 | 45.6 | 6.8 | 47.6 |
| | Gansu | 42.5 | 21.9 | 35.7 | 23.0 | 29.8 | 47.2 |
| | Ningxia | 35.1 | 14.5 | 50.4 | 39.3 | 19.3 | 41.4 |
| | Qinghai | 6.3 | 40.1 | 53.6 | 16.1 | 33.3 | 50.5 |
| Northeast China | Heilongjiang | — | — | — | 7.5 | 46.5 | 46.0 |
| | Jilin | — | — | — | 4.9 | 55.8 | 39.3 |
| | Liaoning | — | — | — | 6.5 | 49.6 | 43.9 |

Note: "—" indicates no data available.

## 4. Discussion

### 4.1. Grey Water Footprint Influencing Factors

By calculating the results, it was found that the main factors that contributed to a high $WF_{grey}$ for wheat in North China were the high leaching nitrogen application rates. This finding is in agreement with the result of high nitrogen application proposed by Zhang et al. [9]. The main factors determining high $WF_{grey}$ in Northwest China were large leaching nitrogen application rates, as shown in Figure 5a.

Further analysis of $WF_{grey}$ and factors influencing it in the relevant provincial administrative regions showed that the high $WF_{grey}$ of wheat in North China was mainly affected by the high $WF_{grey}$ in Inner Mongolia. The nitrogen application rate in Inner Mongolia was the highest in North China, while its yield per unit area was the lowest because of the combined effect of a high nitrogen application rate and low yield while $WF_{grey}$ was twice as high as the North China divisional average (Figure 6). Ningxia had a $WF_{grey}$ higher than the divisional average in Northwest China. The wheat nitrogen application rate of Ningxia was at the Northwest average. However, its yield per unit area was the lowest in the division, which led to the highest $WF_{grey}$ recorded for wheat. The nitrogen application rate and yield per unit area of wheat in Xinjiang were both the highest in the Northwest division. Compared with the nitrogen application rate, yield per unit area was higher than divisional average. The combined effect made $WF_{grey}$ in Xinjiang the lowest in the division (Figure 7). In general, wheat $WF_{grey}$ was high in North and Northwest China, mainly because of high nitrogen application rates and low yield per unit area. It is necessary to limit the nitrogen application rate for wheat specifically in Inner Mongolia and Xinjiang, and to improve yield per unit area in Inner Mongolia, Shanxi, Gansu, and Ningxia.

The main factors responsible for the high $WF_{grey}$ of maize in Southwest China were low yield per unit area and had a high nitrogen application rate. Similarly, the main factors responsible for the high $WF_{grey}$ of maize in Northwest China was the high nitrogen application rate. Maize yield per unit area was higher in the Northwest than the cultivation divisional average, which reduced the extent of increase of $WF_{grey}$ (Figure 5b).

Further analysis of $WF_{grey}$ and the factors influencing it in the relevant provincial administrative regions showed that the highest $WF_{grey}$ of maize in Southwest China was mainly affected by performance in Yunnan and Guizhou. The nitrogen application rate for maize in Yunnan was the highest, whereas its yield per unit area was low in the division. On the other hand, the corresponding rate in Guizhou was close to the divisional average, but its yield per unit area was the lowest in

the division. These combined effects resulted in a high WF$_{grey}$ in the Southwest Division (Figure 8). Fertilizer application rates in maize in the five northwestern provinces were approximately the same. The high WF$_{grey}$ of maize in Northwest China was affected by the low yield per unit area in Shaanxi and Gansu. Shaanxi had the lowest yield per unit area in the division, while Gansu was the second lowest, which makes the WF$_{grey}$ of these two provinces higher (Figure 8). In general, the WF$_{grey}$ of maize was high in Southwest China and Northwest China mainly because of the high nitrogen application rates and the low yields per unit area. Particularly, in the Northwest division, the nitrogen application rate in maize was 35% higher than the cultivation divisional average. It is necessary to limit the nitrogen application rate specifically in Northwest China and Yunnan, and to improve the yield per unit area in Shaanxi, Guizhou, and Gansu.

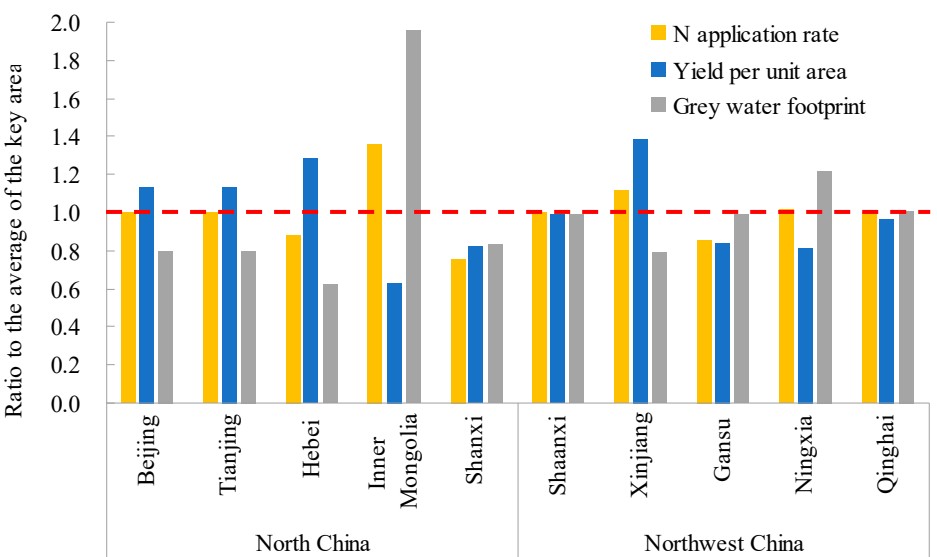

**Figure 7.** Fold-change of the ratio of factors affecting WF$_{grey}$ of wheat production to divisional average in provinces of North China and Northwest China.

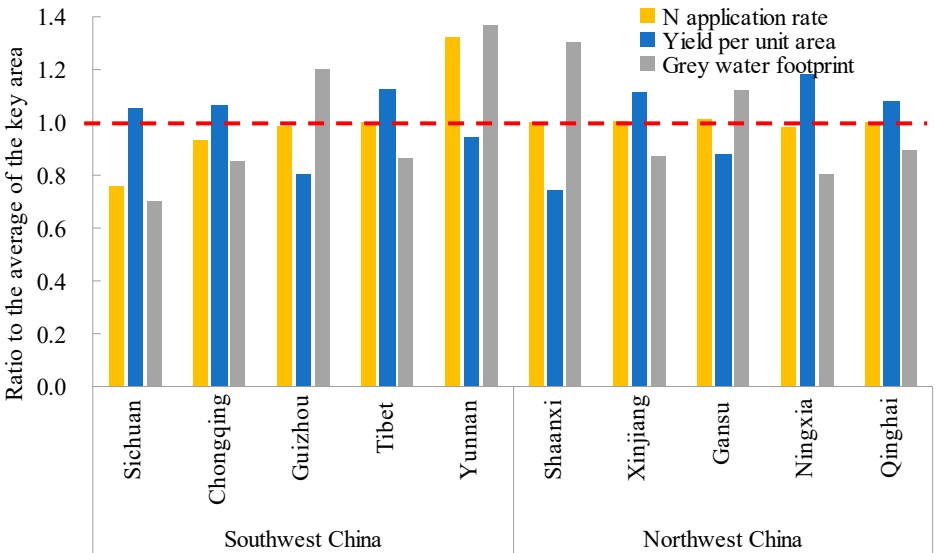

**Figure 8.** The fold-change of the ratio of factors affecting WF$_{grey}$ for maize production to the divisional average in provinces of Southwest China and Northwest China.

*4.2. Total Water Footprint and Grey Water Footprint*

There were significant divisional differences in the case of wheat. North China had the highest wheat $WF_{tot}$ at 2.10 m$^3$/kg, while Southwest China had the lowest at 0.90 m$^3$/kg. In the provincial administrative regions, $WF_{tot}$ was high in Inner Mongolia, Jiangxi, Gansu, Shanxi, and Ningxia (Figure 6a). The regional difference of $WF_{tot}$ of maize was not significant, except for Tibet. The average $WF_{tot}$ in the Southwest division increased to 1.48 m$^3$/kg due to the high $WF_{green}$ in Tibet, while the average of the remaining four divisions was similar (Figure 6b).

As for the ratio of $WF_{grey}$ to $WF_{tot}$, the order of provincial administrative regions with more than 50% $WF_{grey}$ for wheat, from the highest to lowest, were Inner Mongolia, Qinghai, Tianjin, and Ningxia, while, for maize, they were Yunnan, Shaanxi, and Qinghai. The large proportion of $WF_{grey}$ indicates that it takes more water to assimilate the pollution load to the water quality target concentration, which also utilizes a large amount of local freshwater resources. Therefore, the fresh water volume utilized by crop fertilization must be taken into consideration in divisional agricultural planning, for which the fertilization utilization estimation can be calculated backward in the constraints of local water available and environmental capacity. If the estimated fertilization is lower than the formulated fertilization when testing the soil, it will not reach a rational benefit. Further measures, such as a reduction of equivalent cultivated areas, should be taken to minimize their effect on water pollution.

When $WF_{grey}$ and $WF_{blue}$ were compared (Table 4), the former was over 1.3 times higher than the latter in all three divisions, except for the Southwest region, where the fertilizer application rate was low. The growth season for maize falls within the rainy season. Thus, the demand for blue water is relatively small. Most of $WF_{grey}$ was more than two to three times higher than $WF_{blue}$, which indicates that, even for crops such as maize, which generally require less water for irrigation, the water resources utilized and contaminated when fertilizers are applied in excess cannot be ignored. In the five divisions, the ratio of $WF_{grey}$ to $WF_{tot}$ in North China and Northwest China is 53.7%, which is higher than that of $WF_{blue}$ to $WF_{tot}$, i.e. 45.72% while the ratio of $WF_{grey}$ for maize is higher than that of $WF_{blue}$. In these two divisions, in terms of a lack of water resources, the total annual volume of $WF_{grey}$ from 2012 to 2016 in North China and Northwest China were approximately 111% and 39% of current irrigation water utilization. The total agricultural water utilization including $WF_{grey}$ will be higher than the agricultural water available. The ignorance of impact of $WF_{grey}$ in practice is one of factors resulting in less improvement of the environment in a long period. Two appropriate ways to protect the aquatic environment and promote the sustainable utilization of water resources in these two divisions are suggested as: 1) cutting down the current fertilizer utilization by 30% in North China and 12% in Northwest China, and 2) cutting down crop-cultivated areas and enhancing crop yield per unit.

## 5. Conclusions and Implications

1.  The average $WF_{grey}$ of wheat from 2012 to 2016 in China ranked as follows: North China 1.13 > Northwest China 0.74 > East China 0.29 > Southwest China 0.09 m$^3$/kg. $WF_{grey}$ in North and Northwest China were relatively high and showed an overall increasing trend, mainly because of the high nitrogen application rates and the low yield per unit area. It is necessary to limit the nitrogen-application rate for wheat specifically in Inner Mongolia and Xinjiang, and to improve the yield per unit area in Inner Mongolia, Shanxi, Gansu, and Ningxia. The average $WF_{grey}$ of maize ranked as follows: Southwest China 0.56 > Northwest China 0.54 > North China 0.44 > Northeast China 0.37 > East China 0.24 m$^3$/kg. It was relatively high in Southwest China and Northwest China mainly because of the high nitrogen application rates and the low yield per unit area. The nitrogen application rate in the Northwest division was 35% higher than that of the cultivation divisional average. It is necessary to limit the nitrogen application rate for maize specifically in Northwest China and Yunnan, and to improve the yield per unit area in Shaanxi, Guizhou, and Gansu.

2.  As for $WF_{tot}$ of the irrigation area, wheat reached the highest value, with 2.10 m$^3$/kg in North China, followed by Northwest China with 1.61 m$^3$/kg. In the provincial administrative

regions, Inner Mongolia, Jiangxi, Gansu, Shanxi, and Ningxia showed a high $WF_{tot}$ of wheat. The divisional difference in $WF_{tot}$ of maize was not significant. The Southwest showed the highest $WF_{tot}$ with 1.48 m$^3$/kg, which was due to the high $WF_{green}$ in Tibet.

3. In the provincial administrative regions where $WF_{grey}$ accounted for more than 50% of $WF_{tot}$, ranking for wheat from the highest to lowest was as follows: Inner Mongolia, Qinghai, Tianjin, and Ningxia, while, for maize, the corresponding ranking was Yunnan, Shaanxi, and Qinghai. It is necessary to pay attention to and effectively control fertilizer application rates. Except for the Southwestern division, where fertilization is relatively low, the $WF_{grey}$ of wheat was generally 1.3 times higher than that of the $WF_{blue}$, and two to three times higher than for maize in most cases. In the planning and development of agricultural water resources, it is necessary to consider the amount of water utilized for NPS pollution, such as chemical fertilizers and pesticides.

4. The $WF_{grey}$ is an indicator of pollution reflected by the amount of clean water required to assimilate polluting chemicals. The higher the concentration of the pollutants in the receptor water-body, the smaller the difference to the maximum allowable concentration, and the greater the amount of water required to assimilate the discharged chemical. In water deficient areas, the $WF_{grey}$ will undoubtedly increase the water shortage in the region. When analyzing and planning the water demand for agricultural irrigation, it is necessary to pay attention to and analyze the amount of water required for NPS pollution, such as by fertilizers and pesticides. In current evaluations of agricultural water demand planning and water resources development and utilization, focusing on blue water ($WF_{blue}$) only while neglecting $WF_{green}$, and ignoring $WF_{grey}$ could lead to a gross overestimation of the availability of agricultural water resources and failure to meet the goal of sustainable utilization of water resources.

5. This study discussed the effects of nitrogen fertilizer, nitrate-nitrogen leaching rate, and yield per unit area on the magnitude of the grey water footprint of wheat and maize. Because the effects of pesticides and organic fertilizers were not considered and the natural background concentration of the receiving water body is unavailable, the resulting grey water footprint values reported herein are relatively low. It might be underestimated for assessing the negative impact by over-using fertilizer. In-depth studies over an extended duration are warranted in the future.

**Author Contributions:** Conceived the study and writing-review & editing by L.W. Instructed data analysis and writing-original draft by Y.Z. Calculated blue water requirement and analysis by L.J. and G.Y. Date collection and procession by Y.Y. and W.W.

**Funding:** This research was funded by the Chinese Academy of Engineering's consulting project Research on Key Strategic Issues of Agricultural Resources and Environment in China (No. 2016-ZD-10).

**Acknowledgments:** The authors would like to give thanks to Yu LIU and her team, who offered very necessary help on blue and green water calculations.

**Conflicts of Interest:** The authors declare no conflict of interest.

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
