# Peer review of "Spatial Characteristics and Implications of Grey Water Footprint of Major Food Crops in China"

_water, doi:10.3390/w11020220_

Round 1
Reviewer 1 Report
Please see comments in the attachment.
Best regards,
Reviewer

Author Response
Response to Reviewer 1 Comments
(Many thanks to your meticulous comments! We have made a comprehensive revision according to the comments and responded them point by point below. After correcting, we believe that this paper has been improved a lot and expect it acceptable. If there is any further adjustment needed, would you kindly point it out.
Note: 1) The line numbers appear in all Responses here is in the status of using the "Track Changes" function in Microsoft Word. They might appear different in the status of final version. 2) Some corrections in the text are according to the comments from other reviewer(s), would you not mind to disturb you too much.)
Point 1: There is no information about the blue and green water footprint methodology, although the results of these indicators also are presented in the article;
Response 1: This article is describing mainly the calculation of grey water footprint (WFgrey) and its spatial distribution in China. For understanding the effect on environment by WFgrey, we also used the terms of green and blue water footprints. Due to the word number limitation of the journal to articles, we didn’t provide more information for them in last version of the article. We have revised it based on this point of comments by adding the definitions of green and blue water footprints without extending their calculation. The detailed revision is in Line 105 to 109 of the text. We have also moved a relevant sentence describing total water footprint to paragraph 2.1 from 3.3, so that the title of paragraph 2.1 was changed synchronously.
Point 2: The assumed goals have not been fully implemented, i.e. in the Discussion section Authors concluded 1-2 sentences, which are very general; in the style:" The application rates of agricultural fertilizers need to be considered and effectively controlled in these areas to minimize their effect on water pollution" or " Therefore, the control and rational use of fertilizers are extremely important for protecting the aquatic environment and promoting the sustainable utilization of water resources". This text should be refer to the results obtained and specifically to the analyzed region. It may also be worth referring to the good practices that will allow to obtain the assumed technical support for agricultural water planning or prevention and control of NPS pollution;
Response 2: We agree this point of comments and have revised by adding some information to support our conclusions in the Discussion section. This information is referred from the results obtained and specifically to the analyzed region. The detailed revision is also referred in Response 4.
Point 3: What's the source of data for 2012-2016, e.g. for nitrogen application rate per unit area? Based on the data presented in tables 1 and 2, it is difficult to determine the described trends of changes. Perhaps it is worth showing (in the article or in the annex) a table with data for the analyzed period and regions?;
Response 3: All sources of data for 2012-2016 including crop cultivated areas of wheat and maize and nitrogen application rate per unit area are quoted from the references [2] and [40] (renewed as [34]). We accepted this point of comments on describing the trends of changes. We added a figure (renewed as Fig. 1 or Line 132 to 136 in the text) to reflect the relation between nitrogen application rate and leaching rate of wheat and maize and added a relevant table (renewed as Table 3 or Line 153 in the text) to show the trends of changes of nitrogen application from 2012 to 2016. The detailed revision is in Line 116 to 119 and Line 154 to 156 of the text.
Point 4: The discussion section is more continuation of the presentation of results from the Results chapter, than discussion of the result obtained in that paper. Discussion should explore the significance of the results of the work. It should integrate your findings in a comprehensive picture and place them in the context of the existing literature. Please see at the above mentioned comments about that section and guide for Authors at the journal website;
Response 4: By learning "Water -- Instructions for Authors" more carefully, we revised and supplemented the Discussion section by proposing some concrete suggestions based on the discussion on higher grey water footprint in the key provincial administrative regions of North China and Northwest China. The detailed revision is in paragraph 4.2 or Line 329 to 336 and Line 340 to 354 of the text. Furthermore, combined with the Discussion section, we have revised the last paragraph (renewed as conclusion (5)) in the Conclusions and implications section to point out some weakness of the study which highlights limitation of the result application and have mentioned some future research directions. The detailed revision is in Line 392 to 397 of the text.
Point 5: Please check the correctness of writing NO3 – and other ions in the whole text;
Response 5: We appreciate this point of comments and rechecked all similar symbols in the whole text.
Point 6: Page 3: It’s not correct [21, 22, 23, 24, 25, 26, 27, 28, 29, 30, 31, 32, 33, 34, 35, 36, 37, 38]. It will be better [21-38]. Please see instructions for Authors;
Response 6: We appreciate this point of comments and revised it in Line 118 of the text.
Point 7: Page 6: instead of “The Water Footprint Assessment Manual 5 will be better Hoekstra et al. [5], Please see instructions for Authors.;
Response 7: We appreciate this point of comments and revised it in Line 164 of the text.

Reviewer 2 Report
The paper has been dealing with an interesting topic and fits well in the journal. It has, however, some scope for the improvement. The methodology is based on a manual 8 y old now and some other developments should be considered. The Water Footprint shall be based on Life Cycle Assessment considering everything from the cradle to grave and actually based on the most recent developments towards the circular economy from the cradle to the cradle. What is the definition of Greywater and how it was achieved? It is very good, that N has been considered as well, however, a proper way is to include the assessment of Nitrogen Footprint as well. It very much suffers from bulk referencing. Multiple references are of no use for a reader and can substitute even a kind of plagiarism, as sometimes authors are using them without proper studies of all references used. In the case, each reference should be justified by it is used and at least short assessment provided. Other formal issues to be considered: Would you use symbol h for an hour, y for a year, d for a day, s for a second and min for a minute? Would you use SI symbol t for a ton, g for gram, m for a meter, k for thousand and M for a million? For the highlights, you should follow the publisher guidelines. They have to be short and condensed. The list of references needs considerable tiding up. Proper using capitals, same style in presenting family names and given names initials ect. For books, reports, patents, thesis and dissertations the place and country where published should be provided. For papers written in English all titles of references should be fully translated and behind the references, a note e.g. (in French) should be made Would you avoid using "we" style in an archive paper? Using in the list of references just et al is politically and socially incorrect and also impolite. Names of the minimum first six authors have to be provided. For the text clarity would you refrain from using additional words, mostly meaningless filler words, which can be omitted or some archaic words see e.g. “respectively”, “thus”, “hence”, therefore”, “furthermore”, “thereby”, “basically,”, “meanwhile”, ”wherein”, “herein”, “Nonetheless”, “Perceivably,” ,etc. ? Figures as Fig 3 suffers from an incorrectly selected scale. This is a part of BSc studies to select a proper scale the give the figure the right informative value. The paper still needs development and perhaps also some more research.
Author Response
Response to Reviewer 2 Comments
(Many thanks to your meticulous comments! We have made a comprehensive revision according to the comments and responded them point by point below. After correcting, we believe that this paper has been improved a lot and expect it acceptable. If there is any further adjustment needed, would you kindly point it out.
Note: 1) The line numbers appear in all Responses here is in the status of using the "Track Changes" function in Microsoft Word. They might appear different in the status of final version. 2) Some corrections in the text are according to the comments from other reviewer(s), would you not mind to disturb you too much.)
Point 1: The paper has been dealing with an interesting topic and fits well in the journal. It has, however, some scope for the improvement. The methodology is based on a manual 8 y old now and some other developments should be considered. The Water Footprint shall be based on Life Cycle Assessment considering everything from the cradle to grave and actually based on the most recent developments towards the circular economy from the cradle to the cradle.
Response 1: Thanks for this point of comments. We do think it is a very interesting topic, on which we had made our effort, and this article fits well in the journal. What we understood for the life cycle of non-perennial plant or crops like wheat and maize is that they have life length from seeding to harvest. Therefore, the water footprint of wheat and maize being discussed in this article has a whole life cycle from seeding to harvest and they grow within one year.
Point 2: What is the definition of Greywater and how it was achieved? It is very good, that N has been considered as well, however, a proper way is to include the assessment of Nitrogen Footprint as well.
Response 2: The definition of grey water has been described in Introduction section (Line 45 to 47 of the text) and the way to achieve it has been described in formula (1) including the explanation of its relevant parameters (Line 87 to 90 of the text). For reflecting the change of Nitrogen in 2012-2016, a table (renewed as Table 3 or Line 153 of the text) of nitrogen application information of wheat and maize was added.
Point 3: It very much suffers from bulk referencing. Multiple references are of no use for a reader and can substitute even a kind of plagiarism, as sometimes authors are using them without proper studies of all references used. In the case, each reference should be justified by it is used and at least short assessment provided.
Response 3: We admitted that some references were of less use for readers. We omitted some less relevant references and tried to keep all left ones justified and related with the data quoted in the article. Accordingly, the references were renumbered both in the text and in reference list. Additionally, the 76 sets of data, quoted from multiple references, were filtrated out from much more data analyzing by accepting the data with positive relation between nitrogen application rate and leaching rate and in rational ranges and by eliminating others unreasonable and with less relevant references. A figure (renewed as Fig. 1 or in Line 132 to 136 of the text) was added to describe the effectiveness and rationality of data used in the articles. The detailed revision is in Line118 to 119 of the text.
Point 4: Other formal issues to be considered: Would you use symbol h for an hour, y for a year, d for a day, s for a second and min for a minute? Would you use SI symbol t for a ton, g for gram, m for a meter, k for thousand and M for a million? For the highlights, you should follow the publisher guidelines. They have to be short and condensed.
Response 4: We appreciate this point of comments. We corrected all the symbols following the direction in "Water -- Instructions for Authors". For the highlights, we have reorganized them in five bullets to explain the subject of this article precise, short and condensed. The detailed revision is provided through the submission system and shown as follows.
lIncredible growth of agricultural fertilization has affected water bodies in China
lGrey water footprint (WF) reflects how N-fertilizer application effects on water
lGrey WF of wheat and maize takes high ratio in irrigation beyond imagination
lRegional difference of grey WF of wheat in China is much higher than maize
lGrey WF and its impact might be underestimated or even ignored.
Point 5: The list of references needs considerable tiding up. Proper using capitals, same style in presenting family names and given names initials ect. For books, reports, patents, thesis and dissertations the place and country where published should be provided. For papers written in English all titles of references should be fully translated and behind the references, a note e.g. (in French) should be made
Response 5: We appreciate this point of comments. We rechecked and corrected all references one by one based on the directions in "Water -- Instructions for Authors". We have also renumbered them according their adjustment. (cf. Response 3)
Point 6: Would you avoid using "we" style in an archive paper? Using in the list of references just et al is politically and socially incorrect and also impolite. Names of the minimum first six authors have to be provided.
Response 6: We appreciate this point of comments. We have revised some sentences of "we" style. The detailed revision is in Line 12 to 16, 238 to 242 and 340 to 342 of the text. We have also adjusted the name list of reference authors up to all first six if any.
Point 7: For the text clarity would you refrain from using additional words, mostly meaningless filler words, which can be omitted or some archaic words see e.g. “respectively”, “thus”, “hence”, therefore”, “furthermore”, “thereby”, “basically,”, “meanwhile”, ”wherein”, “herein”, “Nonetheless”, “Perceivably,” ,etc. ?
Response 7: We appreciate this point of comments. We have rewritten some sentences to try to avoid these needless words for the text clarity.
Point 8: Figures as Fig 3 suffers from an incorrectly selected scale. This is a part of BSc studies to select a proper scale the give the figure the right informative value. The paper still needs development and perhaps also some more research.
Response 8: Yes, this figure as Fig. 3 (renewed as Fig. 4 or in Line 231 of the text) seems not readable well. It was used for reflecting the change of grey water footprint in spatial and temporal distribution. Due to the non-obviously change in temporal scale and the word number limitation of the journal to articles, we have revised this figure in which temporal distribution was averaged by 5 year values to highlight the difference between grey water footprints for wheat and maize in spatial distribution. Compared with original figure, it seems more clear and beautiful without any information omitted. The detailed revision is in Line217 to 219 of the text.

Round 2
Reviewer 2 Report
The revisions were completed and from my side, the manuscript has been ready to be published.
Author Response
Response to Reviewer 2 Comments (Round 2)
Thanks for your carefully findings of the text. All suggestions have been accepted as bellow.
Point 1: Line 12: mention more specifically which period you mean (instead of saying"in recent decade").
Response 1: Accepted, this sentence has been adjusted as “The estimated effective increase of agricultural fertilizer applied in China by 10.57 million Mtons in recent years from 2006 to 2016 is a crucial factor affecting water environment.”
Point 2: Line 17: replace "and increased in trend" by "with increasing trend"
Response 2: Accepted.
Point 3: Line 19: replace "wheat" by "wheat's"
Response 3: Accepted.
Point 4: Line 42-43: replace "Arjen 43 and Ashok" by "Hoekstra and Chapagain".
Response 4: Accepted.
Point 5: Line 43-44" replace "completed by the grey water footprint working group of the Water Footprint Network" by "elaborated in the Grey Water Footprint Guidelines".
Response 5: Accepted.
Point 6: In ref list: replace current ref [3] by: Franke, N.A., Boyacioglu, H. and Hoekstra, A.Y. (2013) Grey water footprint accounting: Tier 1 supporting guidelines, Value of Water Research Report Series No. 65, UNESCO-IHE, Delft, the Netherlands.
Response 6: Accepted.
